# Analysis of targeted and whole genome sequencing of PacBio HiFi reads for a comprehensive genotyping of gene-proximal and phenotype-associated Variable Number Tandem Repeats

**Sara Javadzadeh**[1], **Aaron Adamson**[2], **Jonghun Park**[1], **Se-Young Jo**[1,3], **Yuan-Chun Ding**[2], **Mehrdad Bakhtiari**[1], **Vikas Bansal**[4*], **Susan L. Neuhausen**[2*], **Vineet Bafna**[1*]

**1** Department of Computer Science and Engineering, University of California San Diego, La Jolla, California, United States of America, **2** Department of Population Sciences, Beckman Research Institute of City of Hope, Duarte, California, United States of America, **3** Department of Biomedical Systems Informatics, Yonsei University College of Medicine, Seoul, South Korea, **4** School of Medicine, University of California, San Diego La Jolla, California, United States of America

* vibansal@ucsd.edu (Vik.B.); sneuhausen@coh.org (S.N.); vbafna@ucsd.edu (Vin.B.)

**Data availability statement:** Code for analysis and figures are available on

## Abstract

Variable Number Tandem repeats (VNTRs) refer to repeating motifs of size greater than five bp. VNTRs are an important source of genetic variation, and have been associated with multiple Mendelian and complex phenotypes. However, the highly repetitive structures require reads to span the region for accurate genotyping. Pacific Biosciences HiFi sequencing spans large regions and is highly accurate but relatively expensive. Therefore, targeted sequencing approaches coupled with long-read sequencing have been proposed to improve efficiency and throughput. In this paper, we systematically explored the trade-off between targeted and whole genome HiFi sequencing for genotyping VNTRs. We curated a set of $10,787$ gene-proximal (G-)VNTRs, and 48 phenotype-associated (P-)VNTRs of interest. Illumina reads only spanned 46% of the G-VNTRs and 71% of P-VNTRs, motivating the use of HiFi sequencing. We performed targeted sequencing with hybridization by designing custom probes for 9,999 VNTRs and sequenced 8 samples using HiFi and Illumina sequencing, followed by adVNTR genotyping. We compared these results against HiFi whole genome sequencing (WGS) data from 28 samples in the Human Pangenome Reference Consortium (HPRC). With the targeted approach only 4,091 (41%) G-VNTRs and only 4 (8%) of P-VNTRs were spanned with at least 15 reads. A smaller subset of 3,579 (36%) G-VNTRs had higher median coverage of at least 63 spanning reads. The spanning behavior was consistent across all 8 samples. Among 5,638 VNTRs with low-coverage (<15), 67% were located within GC-rich regions (>60%). In contrast, the 40X WGS HiFi dataset spanned 98% of all VNTRs and 49 (98%) of P-VNTRs with at least 15 spanning reads, albeit with lower coverage. Spanning reads were sufficient for accurate genotyping in both cases. Our

https://github.com/sara-javadzadeh/vntr_genotyping/ Whole genome sequencing data is publicly available through the Human Pangenome project through the portal https://humanpangenome.org/data/. Targeted sequencing data generated in this study is available on Sequence Read Archive (SRA) with Bioproject number PRJNA1209963.

**Funding:** This work was supported by the National Institutes of Health grants (HG010149, RM1HG011558 and R01GM114362 to Vin. B.; R01-HG010759 to Vik. B.). The funders had no role in study design, data collection and analysis, decision to publish, or preparation of the manuscript.

**Competing interests:** I have read the journal's policy and the authors of this manuscript have the following competing interests: V. Bafna is a co-founder, serves on the scientific advisory board of Boundless Bio, Inc., and Abterra Inc, and holds equity in both companies.

findings demonstrate that targeted sequencing provides consistently high coverage for a small subset of low-GC VNTRs, but WGS is more effective for broad and sufficient sampling of a large number of VNTRs.

## Author summary

Variable Number Tandem Repeats (VNTRs) are DNA regions where short sequences repeat multiple times. They contribute to genetic variation and have been linked to various traits and diseases. However, their repetitive nature makes them difficult to analyze accurately. Recent advances in long-read sequencing have improved the ability to study these complex regions. In this study, we compared two long-read sequencing approaches: targeted sequencing and whole genome sequencing (WGS). Targeted sequencing focuses on specific VNTRs near genes and those linked to diseases, offering a potentially cost-effective alternative to sequencing the entire genome. Our results showed that targeted sequencing provided high coverage for a small subset of VNTRs but failed to capture most of them, particularly in GC-rich regions. In contrast, WGS successfully covered nearly all VNTRs, though with lower read depth. Although targeted sequencing may be useful for detailed analysis of select VNTRs, WGS remains the better choice for a comprehensive understanding of VNTR variation. These findings can help improve genetic studies and disease research.

## 1. Introduction

Tandem repeats (TRs) are complex, repetitive regions of the human genome, characterized by tandem arrays of repeated sequence motifs. TRs have been categorized based on motif length into two classes: 'short' tandem repeats (STRs) when the repeat unit is at most 5-6 bp, and as Variable Number Tandem Repeats (VNTRs) for repeats with longer motifs. We note that different definitions of VNTR have been used in the literature. Eslami Rasekh et al. define VNTRs as TRs that are copy number variant in a population [1]. They use the term "minisatellite VNTRs" to define TR loci with motif length of 7 or greater. In contrast, several others [2–4] define VNTRs as TRs with motif length greater than 6 without requiring evidence of copy number variability. In this paper, we consider VNTRs as all TR loci with a motif length of 6 or more in the reference genome. STRs/VNTRs are among the most polymorphic regions of the human genome and the polymorphisms manifest as changes in the number of repeat units and less frequently as changes in the repeat unit sequence [5]. Other, complex forms of tandem repeat polymorphisms have been visually observed, but not systematically analyzed [6].

With the availability of whole-genome sequence data, several studies have analyzed the variability of VNTR loci in human populations [1,4,7]. To demonstrate the variability of VNTRs across samples, Lu et al. introduced a repeat pangenome graph built on a set of short and long reads sequenced from 19 samples [4]. They found that the graph constructed using 19 individuals is 27% larger than the same graph constructed based on the GRCh38 reference genome [4].

VNTR genotype changes have been implicated in Mendelian diseases such as medullary cystic kidney disease (*MUC1* VNTR) [8,9], type 1 diabetes (*CEL* VNTR [10]), and expression-QTLs [11]. More recently, polymorphisms in tandem repeats - including VNTRs - have been

associated with complex diseases and phenotypes [12]. Margoliash et al. estimated that STRs account for 5.2% to 7.6% of causal variants associated with 44 common blood traits [13]. Similarly, VNTR polymorphisms have been associated with complex diseases like Schizophrenia [14] and phenotypes such as height [15,16]. Non-coding VNTRs in the *TMCO1* and *EIF3H* genes were associated with glaucoma and colorectal cancer respectively [17], and the corresponding VNTRs showed the largest known association among all common human polymorphisms for the respective phenotype. Together, these studies highlight the importance of investigating the association of VNTR polymorphisms with diseases.

Historically, the repetitive nature and complexity of sequences were a barrier to accurate genotyping of TRs and testing their association with phenotypes. The problem is particularly acute for VNTRs, where even a small change in repeat unit count or mutations in motifs could impact the phenotype as in the *MUC1* and *CEL* VNTRs mentioned above. Although large VNTR expansions have been successfully detected with short-reads, [17], it is difficult to precisely detect small changes in repeat counts without reads that span the tandem repeat sequence and part of the flanking region; the repeated motifs make it difficult to accurately map non-spanning reads. The problem is compounded for VNTRs within segmental duplication regions. For example, the 563 bp VNTR in exon 11 of the *CEL* gene is not only too long to be spanned by short-reads, but also occurs within a duplicated *CELP* pseudogene. In these cases, genomic context (i.e. flanking regions) mapped to long spanning reads can provide additional information for reliable mapping, as shown below in results. VNTR genotyping accuracy is therefore greatly enhanced if the supporting reads span the entire VNTR. This is now feasible with the availability of accurate, long-reads, including Pacific Biosciences HiFi [18–20]. Genome in a Bottle Consortium (GIAB) [21] has been an invaluable resource for tandem repeat analysis where English et al. created a benchmark for tandem repeats based on the HG002 sample [22].

Long read whole genome sequencing could be expensive for population scale studies with large numbers of individuals, or for clinical settings where the goal is to interrogate only a few VNTRs. One option is to incorporate VNTR genotyping with targeted sequencing at regions of interest. This approach facilitates a reliable and reproducible sequencing method for a set of selected gene proximal VNTRs (covering 0.06% of the human genome), while minimizing unnecessary sequencing of off-target regions (99.94% of the human genome). Targeted sequencing with hybridization is widely used and has been proven to work effectively and reliably with short next-generation sequencing (NGS) reads [23,24] and PacBio HiFi reads [25] for single nucleotide polymorphism (SNP) genotyping. In contrast, with traditional amplicon-based methods, target enrichment with hybridization has proven to show high uniformity [26,27] across targeted bases, as well as improved enrichment in the repetitive target regions [28]. Another advantage of probe-based target enrichment compared to the amplicon-based approaches is that allelic dropout is not a concern for probe-based enrichment. Gray et al. demonstrated that target enrichment with probes that are 120 bp long are not typically susceptible to allelic dropout as mismatches (up to seven) does not hinder hybridization [29].

Previous studies have effectively found disease causing variants using a long read target enrichment approach such as Miller et al. [30], Nakamichi et al. [31] and Miyatake et al. [32] using adaptive sampling on Oxford Nanopore (ONT) sequencing technology. While Oxford Nanopore (ONT) sequencing is effective in detecting expansions of tandem repeats, it has reduced sequencing accuracy compared to HiFi reads, especially within tandem repeats [32]. Miyatake et al. [32] observed extensive errors caused by incorrect base-calling in ONT reads in specific VNTRs as well as artifact repeats which were not observed in CHM13 reference genome or PacBio HiFi reads. A more recent target enrichment approach with CRISPR/Cas9

has shown promising results in targeting complex regions of the human genome [33]. However, it is not currently scalable to tens of thousands of target regions.

In this paper, we  explored the tradeoffs between targeted sequencing via hybridization and whole genome sequencing for VNTR genotyping. We first created a resource of phenotype associated and gene proximal VNTRs as a curated set of likely clinically relevant VNTRs. We subsequently generated probes for the targeted VNTRs and a hybridization strategy to sequence the targeted VNTRs in 8 samples. We compared our results to whole genome sequencing of 28 samples from the Human Pangenome (HPRC) Project [34], using as metrics, the fraction of VNTRs spanned by long-reads and the accuracy of VNTR genotyping compared to short-reads and trio data when available.

## 2. Results

**Cataloging and characterizing VNTRs.** To investigate long-read based genotyping of known or likely clinically relevant VNTRs, we compiled two lists. First, 10,787 VNTRs were selected based on their proximity to genes. Specifically, these gene proximal ('G-VNTRs') lay within the gene body (introns and exons), in untranslated regions, or up to 500 bp upstream of the transcription start site (TSS), including promoters and other regulatory regions. A second list of 48 phenotype-associated VNTRs ('P-VNTRs') was selected (Fig 1a), based on published association with specific phenotypes. The 48 P-VNTRs are implicated in metabolic diseases, psychiatric disorders, and different cancers (references at S1 Table and locus information at S2 Table). 39 P-VNTRs were pathogenic, while the other 9 could be deemed  non-pathogenic and 20 were shared within G-VNTRs.

### 2.1. Repeat characteristics of selected VNTRs

In total, the GRCh38 reference DNA sequence of 46% of G-VNTRs and 71% of P-VNTRs exceeded 150 bp , and were unlikely to be spanned by short reads (Fig 1b). We found similar trends of increased complexity in terms of motif lengths and number of motifs. Specifically, 71% of P-VNTRs and 57% of G-VNTRs, respectively, had repeat unit lengths ≥ 20 bp (Fig 1d).

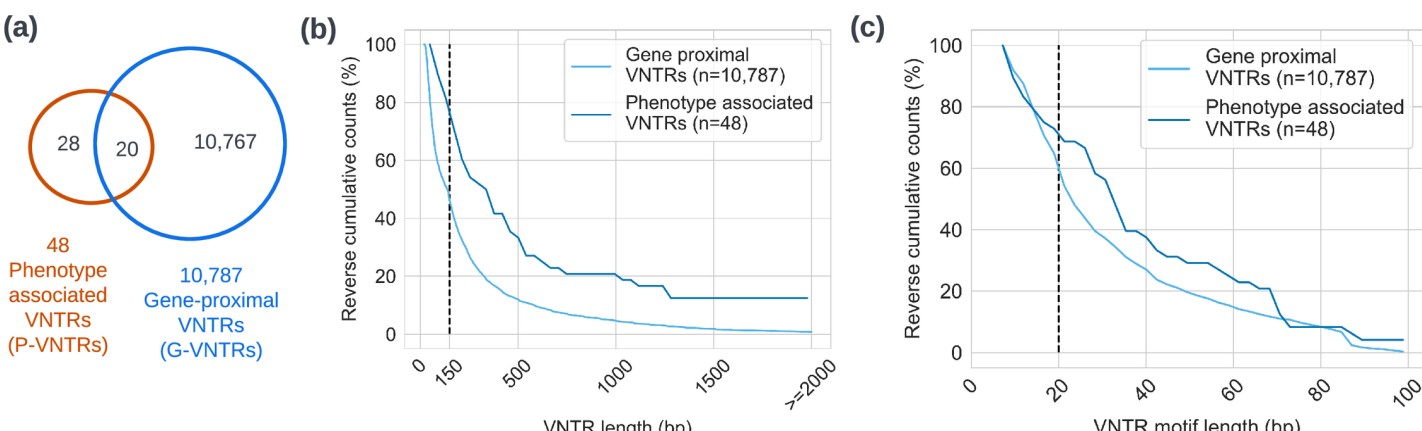

**Fig 1. Targeted VNTRs characteristics and comparison between P-VNTRs and G-VNTRs. (a)** overview of the target VNTR sets including P-VNTRs (Sect 2.5) and G-VNTRs  (Sect 2.1). **(b)**  Percent VNTRs exceeding designated array length in reverse cumulative percentage. A vertical dashed line indicates VNTR length of 150 bp. **(c)** Percent of VNTRs exceeding designated motif length. A vertical dashed line indicates VNTR motif length of 20 bp.

## 2.2. Targeted sequencing consistently performs better at low-GC VNTRs

Briefly, in targeted sequencing with hybridization, probes are designed for a set of target regions and are hybridized with the target DNA fragments. Magnetic beads isolate the hybridized fragments, separating them from non-target sequences. The enriched DNA is amplified via PCR and then sequenced (see Methods section for more details). Out of $10,787$ G-VNTRs, we selected those located on autosomal chromosomes, resulting in $9,999$ G-VNTRs. We designed target probes using the Agilent Sure select platform (Methods), and used those probes to hybridize high molecular weight DNA fragments from 8 samples (S2 Table). The enriched DNA was sequenced using PacBio HiFi, with a median number of 1.5M long-reads per sample and read length of N50=4,208 bp (S4 Table). Notably the shorter read lengths were due to probed selection of fragmented DNA sheared to an average length of 6Kbp (Methods), which was sufficient to span all VNTRs in the study (see S3 Fig).

We observed a large variation in the number of spanning reads per VNTR across different G-VNTRs with the mean of 58 and standard deviation of 86 (Fig 2a). Consistently across the 8 samples, targeted sequencing captured a median of $\geq 15$ spanning reads for $4,091$ (41%) G-VNTRs. Among these $3,579$ (36%) of G-VNTRs had a median spanning coverage $\geq 63$. On the other hand, $3,246$ (32%) of G-VNTRs were not spanned by any read in any sample.

The number of reads was highly homogeneous across all 8 samples for the majority of the G-VNTRs (Fig 2b). The variance in the number of spanning reads across samples was $< 0.2$ in a significant number (35%) of G-VNTRs, after projecting to maximum value 30, the variance of the spanning reads across samples was $< 0.2$ for an increased number (68%) of G-VNTRs. Thus, when a VNTR was well-covered by spanning reads in one sample, it was well-covered in each of the 8 samples, and the coverage depended on the sequence context around the VNTR and the selected probe, rather than the sample.

**2.2.1. Low spanning reads observed in GC-rich regions.** DNA fragments with high GC ($\geq 60\%$) content have difficulties with the PCR amplification step of targeted sequencing. Nearly 65% of the G-VNTRs with 0 spanning reads occurred in regions with high GC-content $\geq 60\%$ (Fig 2a). In contrast, 99% (4,160/4,204) of the well-covered VNTRs (median of at least 15 spanning reads across samples) were located in regions with GC-content $< 60\%$.

While high GC content strongly correlated with poor coverage, nearly 35% of the poorly covered VNTRs were located in regions with low-GC content, suggesting other confounding factors. We noted that probe selection utilizes the VNTR location only as a guideline, and there could be high variability in the distance from a VNTR to the nearest probe sequence. There was a significant drop in the number of spanning reads as the probe distances exceeded 1 kbp (S1 Fig). For VNTRs with minimum distance to the covered region by the probes $\leq 1$ kbp, the decrease in spanning reads was correlated with higher GC quantified by the -0.6 (-0.76 considering logarithmic SRs).

We also tested, but did not find a correlation between the coverage and VNTR lengths (S2 Fig) or VNTR location in repetitive elements. Notably, the targeted sequencing experiments were designed with the G-VNTR lengths in mind (See Methods). In the 8 samples, the median read length was in the range $[4,027 - 4,429]$ bp (S4 Table). For comparison, the 95-th percentile of G-VNTR lengths was $1,341$ (99 percentile: $2,373$, max: $8,938$; also see S3 Fig) which were easily spanned by the targeted reads. While the HiFi whole genome sequences were even longer (median read length in range of $[16–24]$ kbp), the targeted sequence length was sufficient to not impact the coverage of G-VNTRs.

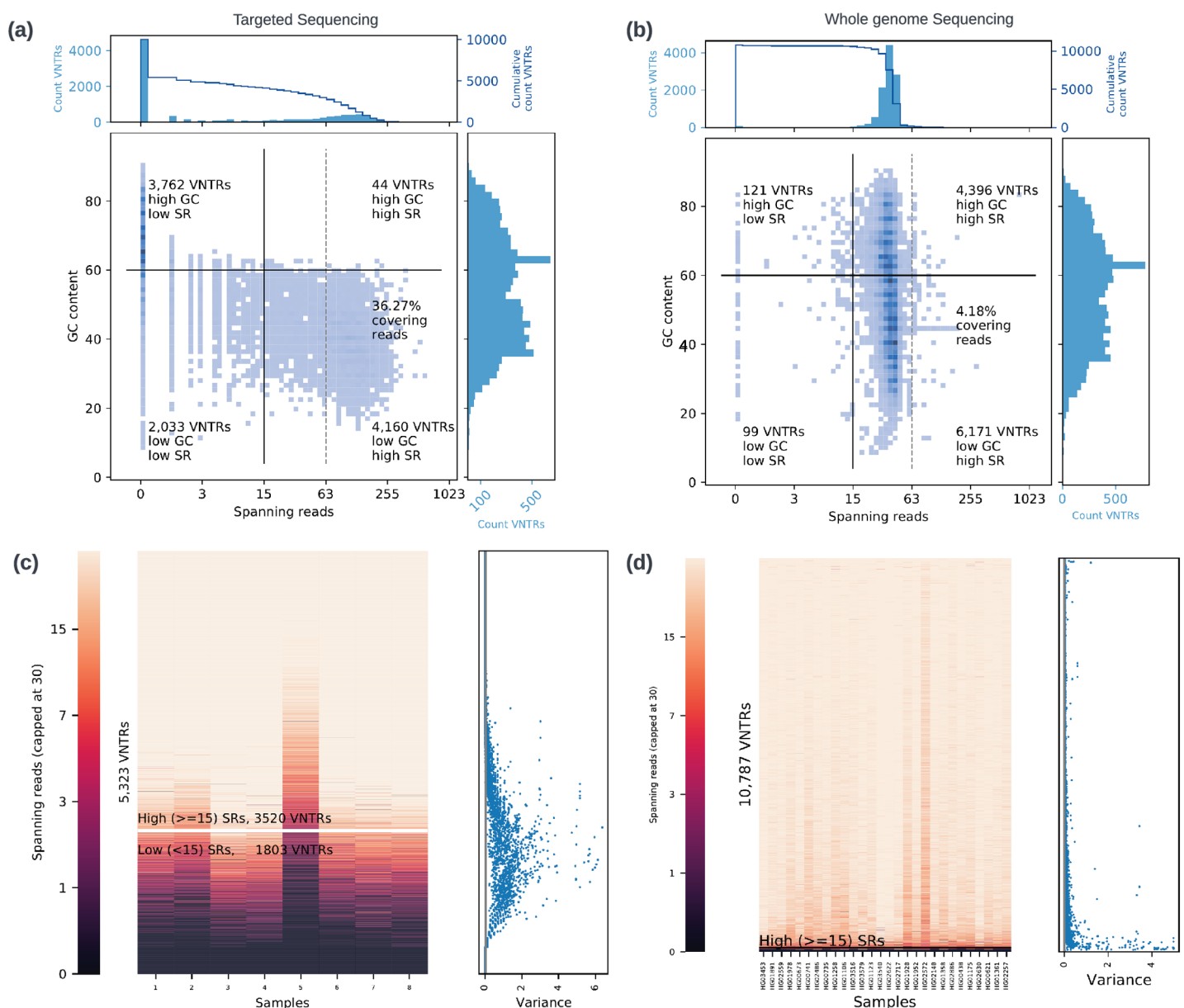

**Fig 2. Comparing spanning reads for target VNTRs in the targeted sequencing (on the left) and whole genome sequencing (on the right) cohorts. (a, b)** GC content and spanning reads for G-VNTRs. Each small square represents the density of VNTRs within that square, the darker the color, the higher the number of VNTRs falling in that square. Vertical and horizontal black lines indicate the thresholds of 15 spanning reads and 60% GC content respectively, with the number of total VNTRs in each quadrant specified on a corner. A dashed vertical line indicates a threshold of 63 spanning reads. The histogram on the top represents spanning reads regardless of the GC content with the outline in navy representing the cumulative counts. Likewise, the histogram on the right represents the GC plot of target VNTRs regardless of the spanning reads. **(c, d)** VNTRs and samples are represented in rows and columns respectively. The brighter the color, the higher the number of spanning reads. The number of spanning reads ≥ 30 are projected to 30. Note the logarithmic scale in the colorbar.

## 2.3. Whole genome sequencing spans VNTRs without bias, but with lower efficiency

We analyzed PacBio HiFi whole genomic data from 28 individuals in the Human pangenome reference consortium project [34]. The mean read coverage across samples after mapping to

the GRCh38 reference was 42x (S5 Table), with 98% of the bases covered by ≥ 15 reads. Correspondingly, 98% of the 10,787 target G-VNTRs were well-covered, i.e. spanned by at least 15 reads (Fig 2c). Once again, the variance in coverage across 8 randomly chosen samples out of the original 28 samples was low (< 0.2) in 95% of the 10,787 G-VNTRs (Fig 2d). Together, the results suggested that whole genome sequencing can be used to genotype VNTRs reliably, and without bias.

However, if VNTR genotyping were the only goal of sequencing, then WGS is not as cost-efficient. Notably, only 4% of reads mapped to G-VNTRs. Targeted sequencing was 9x more efficient with 36% of reads overlapping at least one VNTR region. Moreover, only 156 out of 10,787 (1.4%) of VNTRs were spanned by ≥ 63 reads, in contrast to 3,579 out of 9,999 (36%) of the VNTRs from targeted sequencing.

Thus, while targeted sequencing suffers from PCR-bias, it is much more cost-effective for a subset of VNTRs, once the probes targeting those VNTRs have been demonstrated to be functional.

## 2.4. Genotyping

Due to limited sample availability and the cost of HiFi whole genome sequencing, we did not have samples with both HiFi WGS and targeted sequencing. As a result, a direct comparison was not possible. Furthermore, the true genotypes for our targeted sequencing cohort were not known. Instead, we genotyped G-VNTRs on multiple datasets where validation was possible, to measure the accuracy of genotyping with spanning reads. Although many HiFi specific genotyping tools are available, we use adVNTR [35] which can work for both Illumina and HiFi reads, precisely detecting repeat count variations. Most of our results below would be supported by other genotyping methods.

**2.4.1. Genotyping validation.**   We compared the consistency of genotyping calls between HiFi reads and Illumina reads for the targeted samples. We observed a median of 98.01% consistency (identical genotypes) in VNTRs shared between Illumina and HiFi reads. We observed that 1.98% of VNTRs had *partially consistent* call where one set of reads reported a homozygous site and the other set of reads reported a heterozygous site including the allele in the homozygous call.

63.6% of the partially consistent calls were homozygous in the Illumina reads dataset. We hypothesize that the partially consistent calls were likely caused by the lack of spanning reads on the second allele for the homozygous call. Only 0.14% inconsistent calls were observed among all the targeted VNTRs and samples.

Replicating genotype consistency analysis on the whole genome samples with Illumina and HiFi reads, we observed a similar genotype call consistency of 98.0%. Moreover, 2% of VNTRs had partial consistency (with 82% homozygous calls from the Illumina dataset and 18% homozygous calls from the HiFi dataset), and < 1% of VNTRs had inconsistent calls.

Overall, the comparison between Illumina and HiFi reads revealed that in presence of spanning reads, genotyping with adVNTR is robust in both targeted and whole genome sequencing. Specifically, with 98% identical genotype calls and with the lack of spanning reads being the main reason for non-identical genotypes. Other than the number of spanning reads, long reads were expected to provide more context and fully span the longer alleles that do not fit within the short read boundaries.

**2.4.2. Mendelian consistency.**   To further validate the genotype calls, we computed the Mendelian Consistency among the trios in two cohorts. We obtained Illumina reads for 28 trios in the HPRC dataset and observed over 98.9% Mendelian consistency for 4,984 short VNTRs with spanning Illumina reads. Additionally, we replicated the analysis on the Genome

in a Bottle consortium [36] and we observed 99.7% Mendelian consistency for the HG002, HG003 and HG004 trio using whole genome PacBio HiFi reads, reflecting the robustness of genotyping accuracy using adVNTR on HiFi.

**2.4.3. G-VNTR repeat counts decrease with increased motif length** S4 Fig shows the histogram of repeat counts for the G-VNTRs based on the median repeat count in the WGS cohort. Remarkably, 86% (respectively, 65%) of G-VNTRs had $\leq 20$ (respectively, $\leq 10$) repeat counts among the WGS cohort. However, the length of the repeat unit itself (and therefore, the allele length) was highly variable. We observed a negative correlation between repeat unit counts (RC) and the length of the repeat unit (S4 Fig). As the repeat unit length increased from 6 bp to 20 bp, the median RC dropped from 30 to 10.

## 2.5. Phenotype-associated VNTRs

While the number of phenotype-associated VNTRs is low, we investigated them separately to see if the conclusions matched with the G-VNTRs. Indeed, 71% of P-VNTRs were longer than 150 bp in the GRCh38 human genome (Fig 1c) reiterating the advantage of long and accurate reads in genotyping P-VNTRs. Moreover, 71% of P-VNTRs had repeat unit lengths $\geq 20$ bp(Fig 1d).

Pathogenic alleles in 19% of P-VNTRs involved small changes in repeat motif counts or even single nucleotide variants within the VNTRs (Fig 3a). However, 19 (39%) of 48 P-VNTRs had *high* repeat-unit count (RC) variation, defined as having a difference of at least 10 repeat-units between the phenotype determining and normal alleles in the cohort of study where the association was reported. Another 21 had *low* RC variation, while the remaining 8 VNTRs associated with phenotypes based not on length but on the presence of *specific motifs*, often described by single nucleotide variations within an existing repeat unit (e.g. the *CEL* VNTR in S1 Table). High RC count differences can potentially be detected by algorithms that look for over-representation of specific motifs or oligomers in the VNTRs, and are amenable to discovery by short-read Illumina sequencing [17]. However, low RC differences and/or motif changes are best detected with reads that span the entire VNTR, and these represent 29 (60% of 48) P-VNTRs. Moreover, 71% of these were of length > 150 bp making it difficult to accurately genotype them using Illumina reads.

We further classified the genomic locations of the P-VNTRs (Fig 3a). They were spread across exons, introns, promoters, 3' or 5' untranslated regions (See S2 Table). Within the high RC variation group, slightly more than half (11/19) were located within the intronic region of a gene. Not surprisingly, VNTRs located in exonic regions either had low repeat count variation or had specific motifs contributing to pathology.

**2.5.1. Targeted sequencing captured 8% of P-VNTRs.** Only 4 VNTRs of the 48 P-VNTRs were spanned by $\geq 15$ targeted spanning reads. Notably, 23 (48%) came from high GC content regions, and 17 other VNTRs had probe-distance $\geq 1$ kbp (including one in chromosome X which was not targeted). Finally, 2 VNTRs had lengths 4076 bp and 4506 bp in the hg38 human reference genome, which exceeded the length of the targeted DNA fragments.

**2.5.2. Most P-VNTRs are covered by HiFi whole genome sequencing.** Similar to G-VNTRs, whole genome sequencing covered 92% of P-VNTRs with median spanning coverage $\geq 15$ spanning reads in 28 samples (Fig 3b). *CACNA1C, DUX4* and *ABCA7* VNTRs were not well-covered likely due to VNTR length. The length of the *DUX4* VNTR in the GRCh38 reference genome is 4,506 bp. The *CACNA1C* VNTR length in GRCh38 reference was 319 bp, but that appears to be an underestimate [14]. Lu et al. reported the average length of *CACNA1C* VNTR to be 5,669 bp among 19 genomes [4]. In our cohort, only 2 of 28 samples had at least

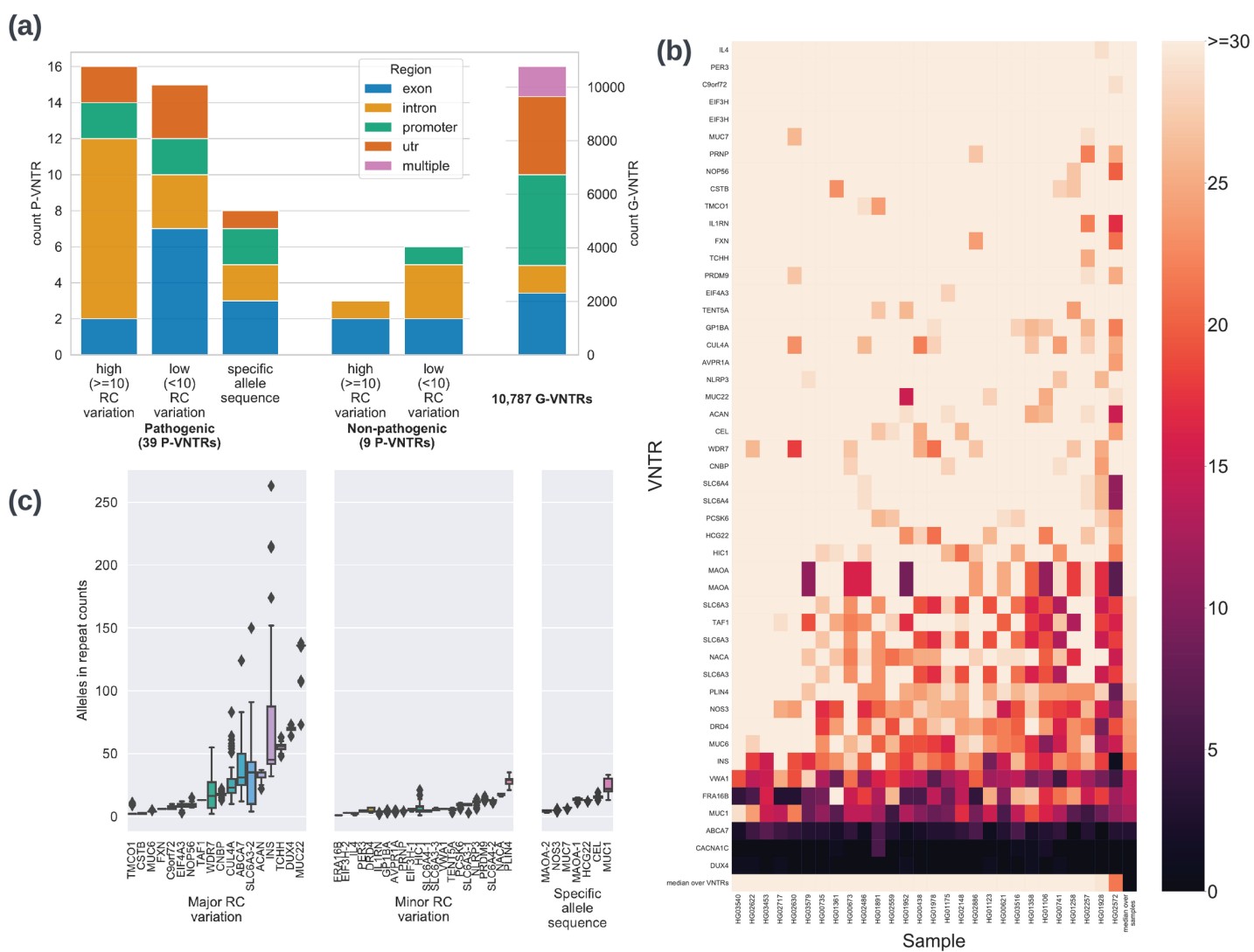

**Fig 3. Spanning reads for P-VNTRs and alleles for all target VNTRs.** (a) breakdown on P-VNTRs based on the polymorphism type and the associated phenotype. The P-VNTRs includes 39 pathogenic and 9 non-pathogenic phenotype-associated loci. Further breakdown of associated polymorphism into high and low repeat count variation with ≥ 10 and < 10 repeat count variation in the cohort under study, and specific allele sequence. A similar region annotation is provided for the G-VNTRs on the right side with a separate scale. For this stacked bar plot, the multiple label indicates VNTRs which span exonic and intronic regions. (b) Spanning reads for P-VNTRs on the WGS cohort. VNTRs and samples are represented in rows and columns respectively. The brighter the color, the higher the number of spanning reads. Spanning reads are normalized by million mapped reads in the sample. (c) Allele distribution of P-VNTRs based on the WGS data on the Pangenome project cohort (28 individuals).

2 spanning reads for *CACNA1C*. With such a limited set of samples, we excluded *CACNA1C* alleles from Fig 3c.

### 2.5.3. Profiling P-VNTR repeat count variations is consistent with the literature.

Fig 3c shows the distribution of repeat unit counts for the P-VNTRs within the whole genome cohort. Because the whole genome cohort is largely unaffected individuals, we do not expect to see large variation in RC counts. Consistent with this, 12 of the 18 high RC variant VNTRs (excluding the *CACNA1C* VNTR) showed a maximum change < 10 in RC count. However, 6 VNTRs showed ≥ 10 units change even in these unaffected individuals. For VNTRs in the low RC variation category, all had < 10 RC difference between any pair of alleles. While the repeat

count variation for VNTRs in the 'specific allele sequence' category were not associated with the respective phenotype, we found small RC variations for all VNTRs within this category.

**2.5.4. Polymorphism in targeted VNTRs.**   While the interesting VNTRs are more likely to be polymorphic in general, five of our P-VNTRs (*FRA16B, VWA1, EIF3H, FXN, TAF1*) were non-polymorphic in the WGS cohort, but are reported to be copy number variant in the literature (see S2 Table). Similarly, we observed that 76% of G-VNTRs were non-polymorphic in the WGS cohort, but a larger fraction may be polymorphic in a larger, more diverse population or among non-healthy individuals. Our conclusions, based on the counts of spanning reads, are likely to hold in most cases.

## 3. Discussion

With the advent of long-read sequencing, including PacBio HiFi, it is increasingly possible to accurately genotype VNTRs, and explore their roles in mediating disease. Accurate genotyping is greatly aided by reads that span the entirety of the VNTR. Here, we compared a targeted sequencing platform, reliant on probe-based enrichment of the VNTR targets against an unbiased (and PCR-free) whole genome sequencing platform. We focused specifically on sets of VNTRs that were either proximal to genes (G-VNTRs) or were previously associated with a phenotype (P-VNTRs).

Our results suggest that a large fraction (nearly 60%) of G-VNTRs could not be effectively targeted and did not yield 15 or larger spanning reads. However, for the VNTRs that could be targeted, we obtained very high coverage (200x average with 41% of the VNTRs covered by at least 15 reads and 36% covered by at least 63 reads). Moreover, the sequencing was efficient in that 36% of the reads sampled a target VNTR. VNTRs in GC-rich regions were very hard to target with the PCR protocols used. In contrast, whole genome sequencing was unbiased with at least 15 spanning reads for 98% of G-VNTRs and 92% of the P-VNTRs. For P-VNTRs, the gap in spanning reads between the two sequencing methods was even more stark, with only 4 VNTRs in targeted sequencing and 45 VNTRs (out of 48) in whole genome sequencing supported by at least 15 spanning reads. However, the coverage was much lower in whole genome sequencing. At 42x average coverage, the efficiency was only 4%, and 1.26% of the VNTRs had spanning read coverage > 63. With some effort for designing functional probes, targeted sequencing could be an appropriate choice for a small subset of VNTRs. However, any exploratory study, where the target VNTRs are not predetermined, is best supported by whole genome sequencing.

An interesting question that arises from our experiments is if we can predict in advance which VNTRs are amenable to targeted analysis. We attempted to address this question using regression methods on a number of features such as VNTR length, GC content, VNTR motif length, and distance to the designed probes. Unfortunately, the data were too sparse to yield insights. With additional data, the ability to identify VNTRs amenable to targeted sequencing would be very useful. Notably, our study identified over 4,000 VNTRs that can be utilized in targeted studies. While our approach focused on VNTRs with clinical relevance, a contrasting approach was followed in a related publication [37], where they focused on a larger set of microsatellites with short motif lengths and deployed a number of filters to enrich for microsatellites that are amenable to targeted sequencing.

The availability of spanning reads is an important consideration for genotyping. We observed 99% Mendelian consistency between Illumina and HiFi reads when testing VNTRs that were spanned by reads. Partial consistency was often correlated with one allele not adequately sampled which was often the case when the Illumina reads did not span the longer

allele. The allele specific coverage should be accounted for in short read genotyping but are less of a concern for long reads.

An important limitation of our study is that we did not have matched samples with both whole genome and targeted HiFi reads to facilitate a direct genotyping comparison. This is due to the limited sample availability and the cost of HiFi whole genome sequencing. Here, we focused first on number of spanning reads to allow for a fair comparison, and second, we used multiple datasets, including Illumina and HiFi reads from the same sample to provide genotyping consistency results.

VNTRs that lie in segmental duplications can be challenging to genotype correctly. For this study, we identified and filtered G-VNTRs that were located in segmental duplications (Methods). However, P-VNTRs were selected based on the literature and four (*PRDM9*, *MUC1*, *CEL*, and *DUX4*) were located in the segmental duplicated regions (see S2 Table). Targeted sequencing was only successful where there were at least 15 spanning reads for the PRDM9 VNTR, for which we verified that the reads correctly mapped to the locus.

In this study, we did not compare differences in stutter errors between targeted and whole genome sequencing. Testing for stutters is typically performed on chromosome X of male samples, where any genotype other than the mode is likely to be a stutter error. Our targeted study samples were used in other studies with Illumina sequencing [38], and no chrX TRs were targeted. It is possible that the accuracy of genotyping targeted VNTRs could be impacted by stutter errors arising during PCR. Published reports on microsatellites with motif length ≤ 4, however, suggest that stutter errors have a strong negative correlation with motif length, and they are uncommon in motifs of length 4 [37]. In contrast, our study exclusively looked at VNTRs where the motif length was at least 6 bp. For longer motifs, stutter errors are unlikely to change motif counts, and AdVNTR is robust to small changes in motif sequences, because it uses underlying Hidden Markov Models (HMMs) allowing insertions and deletions within each motif.

Early results suggest that the availability of long-read HiFi sequencing will yield valuable insights into the allelic distribution of repeat unit counts. For example, our results suggest that many P-VNTRs do not show extreme expansion in the affected individuals, that the number of repeat units is bounded in unaffected individuals even with large variation in total allele length, and accurate genotyping could be impacted by local genomic structure such as segmental duplications. These considerations will become part of future VNTR analyses of the genome.

Long-read sequencing technologies have seen rapid adoption in recent years, leading to an increase in publicly available data that can enhance benchmarking studies. For example, targeted long read sequencing using Oxford Nanopore Technology (ONT) adaptive sampling is another alternative to HiFi reads for genotyping VNTRs. With the advances in ONT adaptive sampling, the target regions can be selected without the need for PCR amplification. Therefore, it is possible to have higher coverage in the high-GC regions where we observed the least number of reads.

## 4. Methods

### Ethics statement

All study participants provided written informed consent for participation in research studies. The study protocol was approved by the City of Hope Institutional Review Board (IRB 09180).

## 4.1. Enrichment and sequencing

**4.1.1. Genomic DNA extraction and fragmentation.** Samples were collected from eight individuals consented and enrolled into an IRB-approved research study (IRB09180). Peripheral blood cell DNA was extracted using a standard phenol chloroform method. 2 μg of DNA from each sample was sheared to a target size of 6 kb using g-TUBEs (Covaris) with RPM of 7000 and spin time of 2 minutes. The fragmented DNA was purified with 0.8X AMPure PB beads (Pacific Biosciences) and eluted in 50 μl Elution Buffer (EB) (Qiagen).

**4.1.2. DNA quantification and quality control.** All DNA samples were quantified using a Qubit 2.0 Fluorometer (Thermo Fisher) and fragment sizes were measured using an Agilent Bioanalyzer DNA 12000 chip (Agilent) following each library production and enrichment step.

**4.1.3. End-repair/A-tailing and adapter ligation.** The sheared DNA for each sample underwent end repair and A-tailing with the KAPA Hyper Prep Kit following the manufacturer's specifications (Roche). Forward and reverse oligonucleotides were annealed to create barcoded universal PacBio adapters (barcoded adapter oligonucleotide pairs bc1001-bc1008 listed S3 Table). Briefly, the corresponding forward and reverse adapter oligonucleotides were combined with 1X Primer Buffer v2 (Pacific Biosciences) in a 20 μl volume to a final concentration of 10 μM. The oligonucleotide pairs were incubated in a thermocycler with the following thermal profile: 80 °C 2 min, 0.1 °C/sec ramp to 25 °C, hold at 4 °C. Next, the end repair and A-tailing reaction products were ligated to 5 μl of 10 μM annealed barcoded adapter in a total volume of 110 μl with an incubation at 20 °C for 30 min. The ligation reaction was purified with 0.5X AMPure PB beads and eluted in 50 μl EB. Two 100 μl PCR reactions were performed on each adapter-ligated product with PacBio universal primer (5' /5Phos/GCAGTCGAACATGTAGCTGACTCAGGTCAC 3') and the Takara LA Taq DNA Hot-Start kit (Takara). We used the following PCR program: 95 °C 2 min, 6 × [95 °C 20 s, 62 °C 15 s, 68 °C 10 min], 68 °C 5 min. The PCR reactions were pooled for each sample followed by purification with 0.5X AMPure PB beads and eluted in 30 μl EB. Following DNA quantification, each sample was diluted to 10 ng/ul with EB. To remove products less than 3 kb, each sample was purified with 3.7X Ampure PB beads diluted to 35 percent (6.5 μl EB, 3.5 μl AMPure PB beads) and eluted with 20 μl EB.

**4.1.4. Target probe design and enrichment.** The target probes were designed using Agilent SureDesign based on the GrCh38 human reference genome. Probes were successfully designed on the 9,999 G-VNTRs in the autosomal chromosomes (see Sect 4.2). We performed targeted enrichment using the SureSelect XT hybridization protocol (SureSelect XT Target Enrichment Manual version D1) with the modifications listed below. Prior to bait capture, indexed samples were quantified with both Qubit and Bioanalyzer DNA 12000 (Agilent) assays to ensure equal representation of each sample in the pool. Equal amounts of each of the 8 indexed samples were pooled to a combined 1.5 μg DNA. PacBio universal primer was added to the pooled samples and dried down to a volume of 4.0 ul. Next, 2.5 μl of both SureSelect Indexing Block 1 and Block 2 were added to the reaction followed by incubation at 95 °C for 5 minutes. Following a hold at 65 °C, The SureSelect hybridization buffer, RNAse block solution, and target probes were added and incubated for 20 h at 65 °C. Following the 20 h incubation, 50 μl Dynabeads MyOne Streptavidin T1 magnetic beads (ThermoFisher Scientific) were added to the reaction followed by wash steps as detailed in the SureSelect XT protocol. Following the wash steps, the magnetic beads were resuspended in 50 μl EB. Two 100 μl post-hybridization PCR reactions were performed with the PacBio universal primer and Takara LA Taq DNA Hot-Start polymerase. We used the following PCR program: 95 °C 2

min, 15 × [95 °C 20 s, 62 °C 15 s, 68 °C 10 min], 68°C 5 min. The PCR reactions were pooled and purified with 0.45X Ampure PB beads and eluted with 50 µl EB.

**4.1.5. SMRTbell library construction.**  DNA damage repair, end-repair/A-tailing, and adapter ligation steps were performed on the captured samples using the SMRTbell Express Template Prep Kit 2.0 (Pacific Biosciences)  following the manufacturer's specifications. The SMRTbell library DNA was purified with 0.5X AMPure PB and eluted with 20 µl EB. Following DNA quantification, each sample was diluted to 10 ng/ul with EB. To remove products less than 3 kb, each sample was purified with 3.7X Ampure PB beads diluted to 35 percent (6.5 µl EB, 3.5 µl AMPure PB beads) and eluted with 20 µl EB. The pool of indexed samples was quantified with both Qubit and Bioanalyzer DNA 12000 resulting in an average fragment size of 5.3 kb.

**4.1.6. Sequencing.**  HiFi sequencing was performed on four SMRT cells with 30 hr movie collection times on a PacBio Sequel II CLR system (Pacific Biosciences) at the City of Hope Integrative Genomics Core. A total of 11 M reads were generated with the median read length of 4.2 kbp and median of 1.64 million reads per sample (Table S4). Demultiplexing was performed and fastq files for the 8 samples generated.

## 4.2. Target VNTR selection

We started with  tandem repeats identified by Tandem Repeat Finder [39] on the GRCh38 human reference genome and limited those to TRs with motif length of at least 6 and excluded VNTRs within LINE and SINE elements. We also excluded VNTRs within the segmental duplicated regions to ensure that the recruited reads were, in fact, from the target region. We then selected the VNTRs within the gene boundaries and up to 500bp of the transcription start site (TSS) which account for the promoter and other regulatory regions. Gene annotations were extracted from the 2019 RefSeq NCBI gene annotations adding up to $10,787$ G-VNTRs. G-VNTRs were subsequently reduced to $9,999$ for autosomal chromosomes.

The database of G-VNTRs and P-VNTRs is available online in the AdVNTR Github directory: https://github.com/mehrdadbakhtiari/adVNTR under "Data Requirements and Pretrained Models (Databases)"  P-VNTRs  were selected from the literature where VNTR-phenotype associations were previously observed. We additionally applied a criterion on the motif lengths of at least 6bp. The S1 Table indicates the VNTR gene name, associated phenotypes, and references.

**4.2.1. Whole genome cohorts.**  In this study we used publicly available data from the HPRC [34] and GIAB [36] cohorts. From the HPRC cohort, we extracted 28 samples with whole genome PacBio HiFi reads already mapped to the human reference genome GRCh38 using Winnowmap [40]. Furthermore, we utilized the whole genome Illumina sequencing reads from the same cohort alongside the whole genome Illumina sequencing reads for the parents of the individuals (when available) for the genotyping validation step (see Sects 2.4 and 4.2). From the GIAB cohort, we used the HG002, HG003 and HG004 trio. Furthermore, for targeted sequencing analysis, we prepared the enriched data on 8 samples.  Detailed information about the City of Hope targeted sequencing cohort (8 samples) and the HPRC cohort (28 samples) including sample ids, coverage and number of mapped reads is provided in S4 and S5 Tables respectively.

**4.2.2. Genotyping.**  We used adVNTR 1.5.0 for genotyping VNTRs with default parameters with the addition of `-pacbio` and `-accuracy-filter` parameters. The `-pacbio` parameter optimizes the genotyping for long reads. Furthermore, `-accuracy-filter` is implemented to discard alleles where little supporting reads are provided. This is to avoid

an erroneous heterozygous call on a homozygous VNTR where the second allele is based on erroneous reads. Quantitatively, prior to inferring the genotype based on a list of repeat counts and corresponding supporting reads, we excluded repeat counts that had less than 3 *reliable* supporting reads. We defined *reliable* supporting reads as reads with at least 10 bp on each of the left and right flanking sides (20 bp total) and a matching rate of at least 95% on each of the flanking regions. Note that the flanking region criteria are applied to reads that are already mapped to the desired VNTR region. In other words, the flanking region criteria are in addition to sequence mapping requirements and are not replacing them. Moreover, for the genotyping validation and Mendelian consistency experiments in this study, one side of the comparison was short Illumina reads. In order to keep the genotyping pipeline and requirement consistent between short and long reads, we enforced a limited 10 bp flanking region criterion.

**4.2.3. Spanning reads threshold.**   To compute a threshold for spanning reads, we estimated the probability of incorrectly calling a heterozygous genotype as homozygous, given $n$ spanning reads. Such an incorrect call in AdVNTR occurs when there are 2 or fewer spanning reads from one allele. Considering a case with $n$ total reads, $x$ reads sampled from one allele, and no allele dropout

$$P_{\text{err}}(n) = P(x \geq n - 2 \text{ OR } x \leq 2) = \sum_{i=n-2}^{n} \binom{n}{i} \cdot 0.5^i \cdot 0.5^{n-i} + \sum_{i=0}^{2} \binom{n}{i} \cdot 0.5^i \cdot 0.5^{n-i} \qquad (1)$$

From Eqn. 1, $P_{\text{err}}(14) = 0.013$, $P_{\text{err}}(15) = 0.0074$. Thus, $n = 15$ is the lowest number of reads where the probability of miscalling a heterozygous genotype is lower than 1%, and it was chosen as the threshold for spanning reads.

**4.2.4. Genotyping validation by Mendelian consistency.**   To validate the genotype call by Mendelian consistency, we used the GIAB trio with whole genome PacBio HiFi reads for HG002, HG003, and HG004. For this analysis, we focused on G-VNTRs. To apply this Mendelian consistency analysis on a wider range of trios, we used the 28 HPRC  trios with whole genome Illumina short reads and with a focus on target G-VNTRs that are < 150bp.

We further filtered out VNTRs that we deemed to be *STR-like* as formally described in the next paragraph. The idea is to discard the VNTRs where the consensus motif has imperfect internal repeat structures e.g. `AAAAAGA` or `TATATCTA`. As these loci are more similar to STRs than VNTRs, we excluded them from MC computations.

**4.2.5. The STR-like filter for VNTRs**  is discarding the VNTRs where the consensus motif has imperfect internal repeat structures. This filter was only applied to the genotype validation process. This is because the genotype of an STR-like VNTR could be incorrectly estimated by the genotyping method and therefore cause an inconsistency where in fact the underlying sequences are consistent in terms of repeat counts. To apply the filter, we computed a score between 0 and 1 for the consensus motif. The premise of the score is to distinguish motifs such as `AAAAGA` and `GGCCTG` by assigning a higher score to the former compared to the latter because it is easier to distinguish motifs in the VNTR corresponding to the second motif compared to distinguishing the motifs in the VNTR corresponding to the first motif. To compute this score, we iteratively masked out characters in the motif and observed if the masked motif included a perfect repeat. For example, masking out the `G` character in `AAAAGA` by a character that could match to any other character (`AAAA-A`), we can find an internal repeat of `AAA` (matches perfectly to `A-A`).

To find an internal repeat, we computed the Hamming distance between the masked motif against a sliding window of itself. In computing the Hamming distance, we matched the

masked character with any other characters i.e. AAA matches with A-A in the example above with a Hamming distance of 0.

Let $M$ be a motif of length $n$, we first compute $M'$ by masking up to $k$ characters Then we concatenate $M'$ with itself and name the concatenated string $M''$. This is to find any possible circular internal repetitions, starting at any index in $M'$. Note that when comparing $M'$ and $M''$, we skip the trivial identity match by starting at the second index and ending at index $n{-}1$.

The normalized score for motif $score(M)$ with length $n$ is computed by finding the maximum similarity ($n-$ Hamming distance) of $M'$ and a rolling window of size $n$ in $M''$ which indicates the window in $M''$ with most similarity with $M'$. A higher score indicates a higher similarity of $M'$ and one of the substrings of length $n$ of $M''$, i.e. AAAAGA gets a higher similarity score compared to AAAGGA.

The value $k$ determining the number of masked characters is decided based on the motif length $n$ to limit the computation time for VNTRs with longer than 40bp motif, we set n to be the $ceil(\frac{n}{10})$ if $n \leq 40$ and 1 otherwise.

Finally, the $score(M)$ is normalized by motif length, therefore the values are between $[0, 1]$ with score of 1 having an internal repeat with least imperfections therefore, more STR-like, and score of 0 indicating that there it is least likely to have internal repeat in the motif. The STR-like filter discards VNTRs with motif score > 0.8.

**4.2.6. Genotyping validation by comparing Illumina and HiFi reads.** When comparing Illumina and HiFi reads for the same sample in the targeted sequencing and HPRC cohorts, we defined three levels of consistency. We call the genotype calls consistent of the repeat count pair derived by AdVNTR from the two read sets are exactly the same. We mark a genotype call partially consistent if there is at least one heterozygous call and there are two unique alleles observed from the two read sets: e.g. if we observe a genotype of $(2, 5)$ based on the HiFi reads and a genotype of $(2, 2)$ on the Illumina reads. This is to consider the chance that the reads supporting the repeat count 5 in the Illumina reads were missing or discarded in the genotyping process due to low number of supporting reads or a short span in the flanking region (see Sect 4.2). Finally, the inconsistent calls cannot be explained by lack of spanning reads. Examples are: a genotype call $(2, 5)$ and $(2, 7)$ or $(2, 2)$ and $(4, 4)$. These examples show an error that is specific to the genotyping tool.

## Supporting information

References for P-VNTRs are provided in S1 Table.
**S1 Table. P-VNTR Genes, associated phenotypes and references.**
(PDF)

**S2 Table. The list of P-VNTRs with region, coordinate, category and pathogenicity information.**
(TSV)

**S3 Table. Barcoded adapter sequences in the targeted sequencing samples.**
(TSV)

**S4 Table. Summary of sequencing read lengths in the targeted sequencing samples.**
(TSV)

**S5 Table. Summary of sequencing read lengths in the whole genome sequencing samples.**
(TSV)

**S1 Fig. Spanning reads as a function of the distance to the closest probe. (a)** The hue shows the GC content percentile for VNTR and 100 bp flanking regions. The black line represents

the trendline computed as a rolling window, averaging the median of 5 consecutive columns. **(b)** the hue represents the density of overlapping points. **(c)** The histogram of spanning reads regardless of the distance to probe. **(d)** The histogram of the distance to probe for the VNTRs with zero spanning reads corresponding the horizontal line on zero spanning reads in (a) and (b) where overlapping points on the horizontal line hides the distance to probe distribution. (TIF)

**S2 Fig. Spanning reads and VNTR length and repeat count in G-VNTRs.** G-VNTRs with GC-content < 60% are plotted here. Spanning reads for each VNTR correspond to the median spanning reads across all samples (in log scale). **(a)** VNTR length up to 99th percentile shown on the X axis. Any VNTR length longer than that is projected to the 99th percentile value. The hue shows the GC content percentile for VNTR and 100 bp flanking regions. **(b)** The X axis is similar to (a). The hue shows the density of overlapping points. **(c)** The X axis indicates VNTR repeat counts instead of VNTR lengths. The hue shows the GC content percentile for VNTR and 100 bp flanking regions. **(d)** The hue shows the density of overlapping points. **(e)** The histogram of spanning reads regardless of the VNTR length. **(f)** The histogram of the VNTR length for the VNTRs with zero spanning reads corresponding the horizontal line on zero spanning reads in (a), (b), (c), and (d) where overlapping points on the horizontal line hides the VNTR length distribution. (TIF)

**S3 Fig. Comparisons of length distributions of G-VNTR, versus reads from targeted and whole genome sequencing.** Read length distribution of a representative sample (id 7) from the targeted sequencing cohort is shown in blue. Similarly, the distribution for a representative sample from the whole genome sequencing cohort (id HG02559) is shown in orange. G-VNTR length distribution is shown with green color. The count in the Y axis is in log scale. In general, whole genome HiFi reads were significantly longer than targeted sequencing. However, both targeted sequencing and whole genome sequencing were long enough to span almost all G-VNTRs. (TIF)

**S4 Fig. Allele distribution for G-VNTRs.** Each VNTR is represented by the mean repeat count across the alleles in the WGS cohort with darker squares representing higher density of VNTRs. The black trend line follows the mean value in each column. The top histogram presents the motif length regardless of motif counts. The right histogram shows VNTR alleles in terms of repeat counts based on the mean allele value in the WGS cohort. (TIF)

## Author contributions

**Conceptualization:** Vikas Bansal, Susan L. Neuhausen, Vineet Bafna.

**Formal analysis:** Sara Javadzadeh, Jonghun Park, Se-Young Jo, Mehrdad Bakhtiari.

**Funding acquisition:** Vikas Bansal, Susan L. Neuhausen, Vineet Bafna.

**Investigation:** Aaron Adamson, Yuan-Chun Ding, Susan L. Neuhausen.

**Methodology:** Sara Javadzadeh, Aaron Adamson, Jonghun Park, Se-Young Jo.

**Resources:** Sara Javadzadeh, Jonghun Park, Se-Young Jo.

**Software:** Sara Javadzadeh.

**Validation:** Aaron Adamson, Jonghun Park, Se-Young Jo.

**Visualization:** Sara Javadzadeh.

**Writing – original draft:** Sara Javadzadeh, Aaron Adamson, Vikas Bansal, Vineet Bafna.

**Writing – review & editing:** Sara Javadzadeh, Aaron Adamson, Jonghun Park, Se-Young Jo, Yuan-Chun Ding, Mehrdad Bakhtiari, Vikas Bansal, Susan L. Neuhausen, Vineet Bafna.

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
