## [Decision Letter · Decision Letter 0]

12 Nov 2024

PCOMPBIOL-D-24-01176Analysis of targeted and whole genome sequencing of PacBio HiFi reads for a comprehensive genotyping of gene-proximal and phenotype-associated Variable Number Tandem RepeatsPLOS Computational Biology Dear Dr. Javadzadeh, Thank you for submitting your manuscript to PLOS Computational Biology. Your manuscript was reviewed by experts who were uniformly enthusiastic about your careful treatment of strengths and limitations of long-read sequencing in VNTR genotyping. After careful consideration, we feel that it has merit but does not fully meet PLOS Computational Biology's publication criteria as it currently stands. Therefore, we invite you to submit a revised version of the manuscript that addresses the points raised during the review process. Please submit your revised manuscript within 30 days Jan 12 2025 11:59PM. If you will need more time than this to complete your revisions, please reply to this message or contact the journal office at ploscompbiol@plos.org. Please include the following items when submitting your revised manuscript:* A rebuttal letter that responds to each point raised by the editor and reviewer(s). You should upload this letter as a separate file labeled 'Response to Reviewers'. This file does not need to include responses to formatting updates and technical items listed in the 'Journal Requirements' section below.* A marked-up copy of your manuscript that highlights changes made to the original version. You should upload this as a separate file labeled 'Revised Manuscript with Track Changes'.* An unmarked version of your revised paper without tracked changes. You should upload this as a separate file labeled 'Manuscript'. If you would like to make changes to your financial disclosure, competing interests statement, or data availability statement, please make these updates within the submission form at the time of resubmission. Guidelines for resubmitting your figure files are available below the reviewer comments at the end of this letter. We look forward to receiving your revised manuscript. Kind regards, Aakrosh RatanGuest EditorPLOS Computational Biology Jian MaSection EditorPLOS Computational Biology

Feilim Mac Gabhann

Editor-in-Chief

PLOS Computational Biology

Jason Papin

Editor-in-Chief

PLOS Computational Biology

 **Additional Editor Comments (if provided):**    **Journal Requirements:****Reviewers' comments:** Reviewer's Responses to Questions

**Comments to the Authors:**

Reviewer #1: The authors present a comparison of a new targeted HiFi sequencing approach to HiFi WGS for VNTRs. Although targeting can provide higher coverage of some targets with much less total reads, they find that most pathogenic VNTRs were not covered due to high GC and/or difficulty designing probes near the VNTR. I think this is a valuable comparison for the community and just have some suggestions to clarify.

1. The first time I read the abstract, I was confused whether the 3rd paragraph was talking about the targeted or WGS approach, so would be useful to start it saying “with the targeted approach”

2. “Specifically, these gene proximal (‘G-VNTRs’) lay within coding regions, 59 untranslated regions, or up to 500 bp upstream of the transcription start site (TSS), including promoter 60 and other regulatory regions. ” Does this include introns?

3. “The enriched 79 DNA was sequenced using PacBio HiFi, with a median number of 1.5M long-reads per sample with 80 N50=4,207 bp ” This is much shorter than HiFi WGS, so I think this is important enough to highlight and perhaps even mention in the abstract

4. “The coverage was bi-modal in log scale 83 with modes at 0 and 2^7 (Figure 2a) “. Maybe replace 2^7 with ~250

5. “Replicating genotype consistency analysis on the whole genome samples with Illumina and HiFi reads, 137 we observed a slightly lower genotype call consistency of 98.0%.” I don’t think this is significantly lower than the 98.01% for targeted, so would be better to say they are similar

6. “Additionally, we replicated the analysis on the Genome in a Bottle consortium29 and we observed 148 99.7% Mendelian consistency for the HG002, HG003 and HG004 trio using whole genome PacBio HiFi 149 reads, reflecting the robustness of genotyping accuracy using adVNTR on HiFi.” Could the authors use the new TR benchmark from GIAB to evaluate accuracy of targeted and WGS calls for HG002? https://www.nature.com/articles/s41587-024-02225-z

7. “However, the length of the repeat unit itself (and therefore, the allele length) was 154 highly variable, suggesting that while polymerase slippage can cause changes in repeat counts, the range 155 of changes remains constrained.” Could the authors explain more how they draw this conclusion?

8. Related to the previous point, could the authors look at noise in the sequencing reads for targeted vs WGS to better understand whether PCR introduces more noise due to slippage?

9. The link to G-VNTRs on github did not work in the pdf

10. I noticed the LPA VNTR was missing from the P-VNTR list. Were very long VNTRs excluded?

11. I could not find a data availability statement for the targeted sequencing

Reviewer #2: I was excited to read this paper, as HiFi targeting approaches for tandem repeats would be incredibly useful and are currently relatively difficult to scale to large numbers of loci. Overall it was an interesting and thorough analysis. Most of my comments are relatively minor.

Major comments:

Can you predict which VNTRs can be targeted? This could help future researchers decide if they should attempt this approach.

Could adjustments to probe design help improve targeted sequencing? If so, what would you recommend? Why was a distance of 1000bp significant do you think?

Did you observe evidence of allelic dropout in this study? If so, at what depth is this a problem?

In section 2.2, specify the fragment size distributions pre- and post-target enrichment and comment on how this impacted read length. Were read lengths (and likely fragment lengths) shorter in the targeted data? Why? How much impact did this have on the number of spanning reads? Was there a particular allele size at which the read length was typically insufficient? This comment is particularly informed by ref 18, which describes using hybridization for HiFi sequencing. Their HiFi read length averaged at 6287 bp (5909–6819 bp, SD = 285 bp). These reads are much shorter than the typical 15kb of WGS. Is this the main reason for few spanning reads? Does this suggest it could work for more modest-sized VNTR loci?

“The coverage of G-VNTRs is tightly distributed on ≤ 20 repeat counts.”

I struggled to understand this paragraph, and couldn’t match it up with the supporting figure. So I’m not entirely sure what point it is making. I suggest editing for clarity.

Could you please elaborate in the results/discussion how much impact PCR had on stutter at these loci?

Data availability statement needed.

Minor comments:

“Variable Number Tandem repeats (VNTRs) refer to repeating motifs of size greater than five bp”

The definition of VNTR here is somewhat non-standard, although I acknowledge that there is uncertainty about this in the literature. Some of the “P-VNTRs” of <= 6bp motifs would be more typically referred to as short tandem repeats (STRs). Two have motifs <5 bp, so defy the introduction here as well.

Examples:

FXN - GAA

CNBP - CAGG

NPO56 - GGCCTG

C9orf72 - GGCCCC

It would probably be more accurate to say the majority of targeted loci were VNTRs with some STRs added, and explain why those were chosen to include as opposed to other longer motif pathogenic STR loci such as BEAN1 (TGGAA), RFC1 (AAGGG), STARD7 (AAATG), TNRC6A (TTTCA), or SAMD12 (TGAAA).

Mention the targeting method in the abstract for additional clarity. One can infer from the term “custom probes”, but better to be explicit.

You could contrast probe design strategy with short read studies, for example:

https://genome.cshlp.org/content/34/7/1008

Fig S1, S2 Looks like overplotting could be an issue, especially at zero. Suggest adding jitter or experimenting with other plot styles that can better show density.

In the results section on the CEL VNTR, please briefly mention why you focus on this locus. Would it be possible to disentangle the pseudogene reads, and if so how? What are the implications for prior/future work on this locus?

“2.2 Targeted Sequencing consistently express low-GC VNTRs”

Seems like there’s a word or two missing here? I think what you meant to is that Targeted Sequencing consistently performs better at low-GC VNTRs?

Section 2.2. Briefly summarize the steps in the target enrichment at the start of the results when it is first introduced. This will give readers to context to understand the relevance of PCR and probe design when it is mentioned later.

Genotyping Validation: I would expect long-read methods to be more accurate than short read here. If you agree, I think you should set this expectation (or explain why it might not be the case). For example, inconsistency could be allelic dropout, or an error in the short-read analysis.

“Notably, 19% of P-VNTRs involved small changes in repeat motif counts” Typo? Should be something like “pathogenic alleles at P-VNTRs differed by a small change”?

“Notably, only 4 VNTRs of the 48 P-VNTRs” and “Notably, 23 (48%) came from high GC content regions”. Are both equally notable? Suggest rephrasing to emphasize the key point.

Why did the samples chosen differ between the targeted and WGS approaches? This should be addressed as a limitation, as if some samples overlapped it would be easier to compare directly between sequencing approaches.

Reviewer #3: Major comments

The Introduction contains no discussion of previous efforts to detect/genotype VNTRs using whole genome sequencing or targeted sequencing. By "detect" here, I mean determining that the tandem repeats are actually copy number variant. These have been done by several groups. Please include a summary of this work.

There is no discussion of testing the tandem repeat reference set to determine if any of the repeats have closely related sequences elsewhere in the genome. Since this is a common occurrence (one example of such an occurrence, the CEL gene VNTR, is mentioned by the authors) and can confound genotyping, this should be carried out and described. If the targeting protocol can be guaranteed to produce unique primers that will not bind to other loci, including segmental duplications, this should also be reported.

Please describe how many of the genotyped G-VNTRs were found to be heterozygous in at least one genome, in more than one genome. That is, are they actually VNTRs (i.e., copy number variant)?

What was the distribution of allele motif copy numbers for all the G-VNTRs. This is not addressed by Figure S2, which groups TRs by motif length.

Minor comments/editing suggestions

Below I list editing comments and questions. Each part is preceded by the relevant page number/line number. My comments are preceded by >>.

--- Abstract ---

Variable Number Tandem repeats (VNTRs) refer to repeating motifs of size greater than five bp.

>>This is an inaccurate statement. VNTRs refer to tandem repeats that are variable in copy number. Not all repeats with motifs size greater than five are variable in copy number. The class of tandem repeats you describe are called minisatellites. Please rephrase and change the wording throughout the manuscript.

with at least 15 spanning reads, albeit with lower coverage.

>>Why is 15 the criterion?

--- Intro ---

1/3

Functionally, TRs have been classified as ‘short’ tandem repeats (STRs), when the repeat unit is at most 5 bp, and as Variable Number Tandem Repeats (VNTRs) when 4 the repeat unit is at least 6 bp.

>>This is an inaccurate statement. VNTRs refer to tandem repeats that are variable in copy number. Not all TRs with motifs size greater than five are variable in copy number. The class of tandem repeats you describe are called minisatellites. Please rephrase and change the wording throughout the manuscript.

2/25

The problem is compounded for VNTRs within low copy or segmental duplication regions.

>>clarify "low copy" here

2/49

we systematically explored the tradeoffs between targeted sequencing via hybridization and whole genome sequencing for VNTR genotyping

>>The manuscript describes one moderately effective targeting protocol and one WGS data set. This does not seem to be systematic.

--- Methods ---

Various places on page 12:

at the recommended settings

using the recommended conditions

etc.

>>these phrases appear in various places in the DNA preparation steps. Please give references to where these recommendations are detailed.

13/311

We started with 10, 787 gene proximal VNTRs (G-VNTRs) which are defined by VNTRs within the gene boundaries and up to 500bp of the transcription start site (TSS) which account for promoter and other regulatory regions.

>>How was the original list of TRs obtained? You describe a selection or filtering process to obtain the G-VNTRs, so there must have been an original list. Where did it come from? Please describe and include a citation.

>>Was there prior evidence that the G-VNTRs were actually copy number variant?

>>Please detail how the downstream "gene boundary" is defined

>>Regulatory regions for genes could be further away than 500bp and 25 of your P-VNTRs are not in this list, so presumably further than 500 bp away from some gene. Why was 500 bp chosen? Please justify.

13/315

The database of G-VNTRs is available online at the AdVNTR Github directory.

>>What is the file name for the G-VNTR set in that directory? I tried looking but couldn't determine the correct file.

13/317

whole genome PacBio HiFi reads already mapped to the human reads using Winnowmap

>>Is "human reads" a typo?

13/318

Furthermore, we utilized the whole genome Illumina sequencing reads from the same cohort alongside the parent short read data for the genotyping validation step (see Sections 2.4 and 4.2).

>>What is the parent short read data?

13/322

However, one sample with id 40431 was excluded from analysis due to low coverage across all targeted regions.

>>Does this mean you used only 7 samples, not 8 in the remainder of the manuscript? You consistently refer to 8. Please clarify.

>>from table S4, sample 40566 had low overall coverage (about 20% of the median across all samples). Did this one have sufficient coverage across the targeted regions? Please define "low coverage."

13/323

Detailed information about the COH targeted sequencing cohort

>>If the abbreviation means City of Hope, please write it out in full or define the abbreviation earlier.

13/332

We define reliable supporting reads as reads with at least 95% matching rate in the flanking region and at least 10bp on each of the left and right flanking sides.

>>Just to be clear, is the minimum required flanking region 10bp on each side (the phrasing is not quite clear on this) and does the 95% match apply to this combined minimum 20 bp?

>>This is important because the introduction stressed the importance of using flanking sequence to distinguish similar repeats occurring in different regions of the genome, to avoid mischaracterization of genotype. 10 bp is not long enough to distinguish similar repeats in different regions and does not take advantage of long reads extended coverage of flanking regions. Here's the relevant sentence from the Introduction (2/26):

For example, the 563 bp VNTR in exon 11 of the CEL gene is not only too long to be spanned by short-reads, but also occurs within a duplicated CELP pseudogene. In these cases, genomic context (i.e. flanking regions) mapped to long spanning reads can provide additional information for reliable mapping

>>As an additional point, was the collection of target TRs screened to determine if they had homologs in other parts of the genome that might cause mischaracterization of genotype? If so, please describe. If not, please justify why not.

13/336

For this analysis, we focused on G-VNTRs. To apply this Mendelian consistency analysis

>>Why was validation by Mendelian consistency limited to the G-VNTR group instead of the P-VNTR group as well?

13/343

The STR-like filter for VNTRs

>>When was this filter applied, only for validation or for the entire analysis described in the manuscript? If only for validation why only then? If for the entire analysis, describe when it was applied.

>>The whole section on STR-like filtering is written in a confusing way and should be rewritten for clarity.

13/344

Manhattan distance

14/352

finding the maximum Hamming distance

>>Please clarify which metric is being used and if Manhattan, define in this context. If Hamming distance, wouldn't you want the minimum Hamming distance?

14/358

Finally, the score(M) is normalized by motif length, therefore the values are between [0,1] with score of 1 having an internal repeat with least imperfections

>>Wouldn't a score of 0 have least imperfections?

13/336

To apply this Mendelian consistency analysis on a wider range of trios, we used the HPRC dataset trios with whole genome Illumina short reads and with a focus on target G-VNTRs that are < 150bp.

>>How many trios is this?

14/356

we set n to be the ceil(n/10 if n/leq40 and 1 otherwise.

>>Please reformat for correction and clarity

>>The P-VNTR group was not described in the Methods.

--- Results ---

4/69

We found similar trends of increased complexity in terms of motif lengths and number of motifs. Specifically, 71% of P-VNTRs and 57% of G-VNTRs, respectively, had repeat unit lengths ≥ 20 bp (Figure 1d), suggesting a more complex motif structure due to an increasing number of motifs with single nucleotide variations.

>>Motif length does not imply complexity. I suggest just stating that many motifs were longer than a certain size. Do you report on the number of motifs with SNPs? If not, drop this comment.

4/72

In contrast with short-read sequencing, PacBio HiFi reads have lengths 10-25 kbp and accuracy higher than 99%, and are capable of spanning all the VNTRs in this study. However, G-VNTRs contribute to only 0.06% of the human genome, motivating the need for more efficient, targeted sequencing.

>>This belongs in the Introduction and I believe is already mentioned there.

4/77

We selected 9,999 G-VNTRs out of 10,787 G-VNTRs that were located on autosomal chromosomes.

>>Does this mean that the unselected G-VNTRs were on the X and Y chromosomes? Or were there other filtering criteria?

4/80

with N50=4,207 bp (Supplemental Table S2).

Does N50 refer to the fragment lengths? State clearly. Also, information on average number of reads is in S4, not S2.

4/83

The coverage was bi-modal in log scale with modes at 0 and 2^7 (Figure 2a).

>>I don't see a number as large as 2^7 for spanning reads in Figure 2.

4/89

The variance in the number of spanning reads across samples was < 0.2 in a majority (35%) of G-VNTRs, after projecting to maximum value 30, the variance of the spanning reads across samples was < 0.2 for an increased number (68%) of G-VNTRs.

>>35% is not a majority

>>Why was this "projecting to maximum value 30" done? It seems obvious that if the top number is reduced, the variance would be reduced because you're discarding the variance caused by higher numbers.

4/97

In contrast, 99% (4,160/4,204) of the well-covered VNTRs were located in regions with GC-content < 60%.

>>how is "well-covered" defined?

6/178

We further classified the genomic locations of the P-VNTRs (Figure 3a). They were spread across exons, introns, promoters, 3’ or 5’ untranslated regions.

>>Is there a table of these classifications?

7/228

Our results suggest that a large fraction (nearly 60%) of G-VNTRs could not be effectively targeted and did not yield any product.

>>At 4/86, you stated that (32%) of G-VNTRs were not spanned by any read in any sample. Does the 60% include those with less than 15 spanning reads?

--- Figures ---

Figure 1

Captions b) c) use of word "Complexity"

>>please use:

"b) Fraction of VNTRs exceeding designated array length"

"c) Fraction of VNTRs exceeding designated motif length"

Figure 2

What type of scale is used for the coloring in parts (c) and (d)? It's not linear and does not appear to be logarithmic.

S2

The hue shows the GC content percentile for VNTR and 100bp flanking regions.

>>why are you showing the GC content only around the VNTR region? Are other references to GC content in the manuscript to this narrow region or to the entire read length?

--- References ---

Not all journal titles are correctly italicized.

Reviewer #4: In this study, the authors present results from comprehensive genotyping of gene-proximal and phenotype-associated VNTRs using targeted Illumina and PacBio HiFi sequencing, and they compare these results with whole-genome PacBio HiFi data. The authors demonstrate good coverage of most VNTR loci of interest (except for GC-rich VNTRs) in their targeted analysis, and report strong Mendelian inheritance consistency between Illumina and PacBio HiFi data. The manuscript is well-written, and the data is presented concisely and neatly. Given the interest in understanding the roles of VNTRs in disease, this paper could make a valuable contribution to the field. For genotyping, the authors used adVNTR, a tool developed by their group, justifying this choice by stating that adVNTR is compatible with both Illumina and HiFi datasets. The reviewer has a few questions and suggestions for the authors to consider, along with minor comments listed below.

1. Please italicize CACNA1C on page 6. There are instances where the gene name is italicized and others where it is not. Consistency would improve readability.

2. There is more data available in the HPRC. Could the authors clarify why they limited their study to 28 individuals?

3. Although the authors mention the reference sequence used, the version is not specified when first mentioned in line 68. Could they please add the reference genome version?

4. While the authors explain why HiFi sequencing data is preferable over ONT, it would be helpful to include a comparison of genotype consistency among Illumina, ONT, and PacBio HiFi. ONT’s adaptive sampling feature eliminates the need for PCR, reducing PCR-associated biases (such as low coverage in GC-rich regions) and additional enrichment strategies, while also providing methylation information. ONT has made substantial advancements, and datasets based on ONT could potentially support VNTR genotyping, if not motif composition analysis.

5. In line with the last comment, did the authors observe consistency in motif composition for VNTRs spanned by reads in the targeted Illumina and HiFi sequencing data?

6. In the trio analysis, where the authors discuss Mendelian consistency, it would be interesting to know if they observed any de novo changes in VNTR motif compositions within the trios.

7. In the abstract, the authors mention that “Illumina reads span only 46% of the 10,787 (G-)VNTRs and 71% of 48 phenotype-associated (P-)VNTRs.” It’s unclear if these are findings from the current study or general knowledge. If these are study results, the reviewer suggests moving this statement to the last paragraph where the findings are discussed. Otherwise, consider rephrasing to clarify that this is known information.

**Have the authors made all data and (if applicable) computational code underlying the findings in their manuscript fully available?**

Reviewer #1: **No: **I could not find a data availability statement for the targeted sequencing

Reviewer #2: **No: **Data availability statement missing. Code present.

Reviewer #3: Yes

Reviewer #4: Yes

PLOS authors have the option to publish the peer review history of their article (what does this mean?). If published, this will include your full peer review and any attached files.

Reviewer #1: No

Reviewer #2: No

Reviewer #3: No

Reviewer #4: No

---

## [Editor Report · Decision Letter 1]

17 Feb 2025

Dear PhD candidate student Javadzadeh,

We are pleased to inform you that your manuscript 'Analysis of targeted and whole genome sequencing of PacBio HiFi reads for a comprehensive genotyping of gene-proximal and phenotype-associated Variable Number Tandem Repeats' has been provisionally accepted for publication in PLOS Computational Biology.

Best regards,

Aakrosh Ratan

Guest Editor

PLOS Computational Biology

Jian Ma

Section Editor

PLOS Computational Biology

---

## [Editor Report · Acceptance letter]

PCOMPBIOL-D-24-01176R1

Analysis of targeted and whole genome sequencing of PacBio HiFi reads for a comprehensive genotyping of gene-proximal and phenotype-associated Variable Number Tandem Repeats

Dear Dr Javadzadeh,

I am pleased to inform you that your manuscript has been formally accepted for publication in PLOS Computational Biology. Your manuscript is now with our production department and you will be notified of the publication date in due course.

With kind regards,

Anita Estes
